# Elevation, disturbance, and forest type drive the occurrence of a specialist arboreal folivore

**David B. Lindenmayer** [ID] [1]*, **Lachlan McBurney**[1], **Wade Blanchard**[1], **Karen Marsh**[2], **Elle Bowd**[1], **Darcy Watchorn**[1,3], **Chris Taylor**[1], **Kara Youngentob**[1]

1 Fenner School of Environment and Society, The Australian National University, Canberra, ACT, Australia, 2 Research School of Biology, The Australian National University, Canberra, ACT, Australia, 3 Centre for Integrative Ecology, School of Life and Environmental Sciences, Deakin University, Geelong, Victoria, Australia

* David.Lindenmayer@anu.edu.au

**Data Availability Statement:** The underlying data has been uploaded to https://doi.org/10.5061/dryad.xgxd254jf.

## Abstract

Quantifying the factors associated with the presence and abundance of species is critical for conservation. Here, we quantify the factors associated with the occurrence of the Southern Greater Glider in the forests of the Central Highlands of Victoria, south-eastern Australia. We gathered counts of animals along transects and constructed models of the probability of absence, and then the abundance if animals were present (conditional abundance), based on species' associations with forest type, forest age, the abundance of denning sites in large old hollow-bearing trees, climatic conditions, and vegetation density. We found evidence of forest type effects, with animals being extremely uncommon in Alpine Ash and Shining Gum forest. In Mountain Ash forest, we found a negative relationship between the abundance of hollow-bearing trees and the probability of Southern Greater Glider absence. We also found a forest age effect, with the Southern Greater Glider completely absent from the youngest sites that were subject to a high-severity, stand-replacing wildfire in 2009. The best fitting conditional abundance model for the Southern Greater Glider included a strong positive effect of elevation; the species was more abundant in Mountain Ash forests at higher elevations. Our study highlights the importance of sites with large old hollow-bearing trees for the Southern Greater Glider, although such trees are in rapid decline in Mountain Ash forests. The influence of elevation on conditional abundance suggests that areas at higher elevations will be increasingly important for the conservation of the species, except where Mountain Ash forest is replaced by different tree species that may be unsuitable for the Southern Greater Glider.

## Introduction

Quantifying the factors influencing the distribution and abundance of plants and animals has long been an important part of ecology [1,2]. This is especially true in regard to threatened species, where it is critical to determine which targeted conservation management interventions are most appropriate to implement and where [3,4]. However, work on the distribution and

**Funding:** This work has been funded by the Victorian Government Department of Environment, Land, Water and Planning. The funders had no role in study design, data collection and analysis, decision to publish, or preparation of the manuscript.

**Competing interests:** The authors have declared that no competing interests exist.

abundance of some species can be particularly challenging because factors at multiple spatial and temporal scales can influence their occurrence [5–8].

The challenges of working on species distribution and abundance have been particularly prominent in cryptic, nocturnal species such as Australian arboreal marsupials [9]. Yet, such work is critical because of the sensitivity of many of these species to disturbances such as wildfires [10], logging [11], and land clearing [12]. Arboreal marsupials also may be sensitive to other drivers such as climate change [13] and the loss of key elements of stand structure, including the abundance of large old trees that many species use for nesting and denning [14].

The Southern Greater Glider is a species of arboreal marsupial of significant conservation concern (*Petauroides volans*) (*sensu* [15]). The species has suffered local extinction in some jurisdictions (e.g. [16]) and major declines in others [17,18]. The Southern Greater Glider is currently being considered for uplisting to endangered. Therefore, an improved understanding of the factors influencing where it occurs is critical for informed assessments of its conservation status and future management. The Southern Greater Glider is a specialist folivore with a diet comprised almost exclusively of eucalypt leaves [19,20]. The species is dependent on hollows for shelter that can take over 100 years to form in trees [21–24]. The species main predators are large forest owls [25], which are themselves of conservation concern [26]. The Southern Greater Gliders is negatively affected by logging, land clearing and wildfire [12,14], although the species has also declined in places where these stressors are absent. The Greater Glider is also known to be heat sensitive [27] and therefore at risk of the effects of climate change, such as increasingly warm overnight temperatures when animals are actively foraging [13,28]. Given these numerous threats and rapidly declining populations, a better understanding of the factors that influence the abundance and distribution of the Southern Greater Glider is essential for its conservation.

In this study, we posed the overarching question: *What factors are associated with the presence and abundance of the Southern Greater Glider in the montane ash forests of the Central Highlands of Victoria, south-eastern Australia.* We constructed statistical models of the presence and conditional abundance of the species in response to ecologically meaningful variables, including forest age, forest type (dominant eucalypt), the abundance of nesting and denning sites (as reflected by a count of the number of large old hollow-bearing trees), and climatic conditions.

## Methods

### Study area

We focused this study on the Mountain Ash (*Eucalyptus regnans*), Alpine Ash (*E. delegatensis*) and Shining Gum (*E. nitens*) forest ecosystems in the Central Highlands of Victoria, south-eastern Australia (Fig 1). Forests dominated by these tree species are collectively termed montane ash forest. The Central Highlands region of Victoria is located approximately 120 km north-east of the city of Melbourne and covers approximately 1/2 degree of latitude and one degree of longitude (37.82S to 37.86S and 145.83E to 146.02E) (Fig 1). The region experiences mild, humid winters with occasional periods of snow. Summers are generally cool with sensor data gathered in 2019–2020 indicating that median daytime temperatures vary from approximately 9.2°C to 11°C and maximum daytime temperatures vary from 39.7°C to 45.8°C (depending on the age of the forest) (Lindenmayer et al., unpublished data).

We have established 183 long-term monitoring sites, each measuring 1 ha, in the montane ash forests of the Central Highlands of Victoria. These long-term field sites encompass a wide range of environmental conditions including the age of stands (since logging or fire), slope, and aspect.

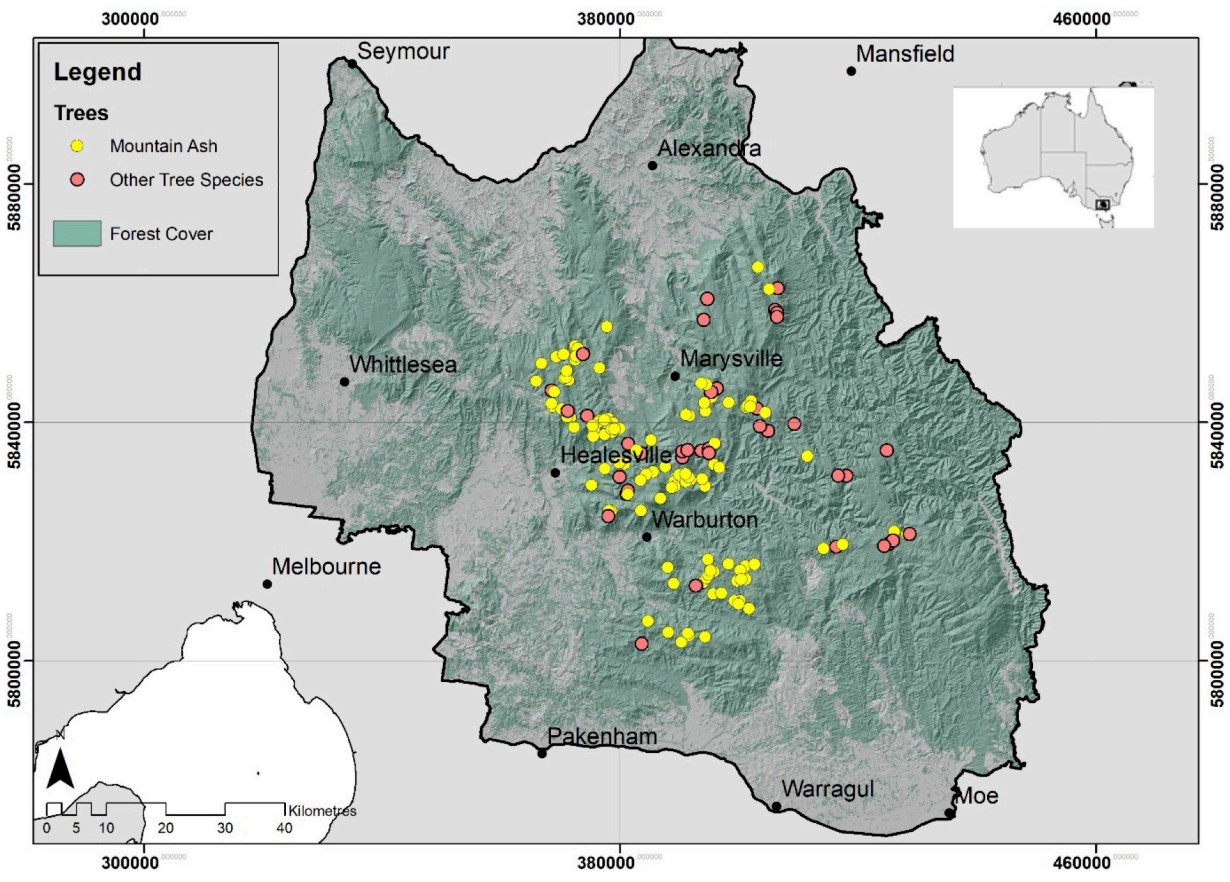

**Fig 1. The location of study sites that were surveyed by spotlighting in the Central Highlands of Victoria.**

The majority of dominant montane ash tree species are obligate seeders that are typically killed by wildfire [29]. They are able to regenerate from canopy-stored seed [30], often producing even-aged cohorts of trees. In this study, we surveyed sites in four stand age classes: **(1)** forest which was burnt in the 2009 wildfires and has regenerated since then (i.e. 12 year old regrowth), **(2)** forest which regenerated after logging or fire between 1960 and 1990, **(3)** forest which regenerated after the 1926–1939 wildfires, and **(4)** old growth forest (i.e. > 120 years since disturbance). We documented the age of the forest and the forest type (Mountain Ash vs Alpine Ash vs Shining Gum) at each site from field-based reconnaissance and disturbance maps from the region generated by the Government of Victoria (see [17]).

## Field surveys of the Southern Greater Glider

We completed spotlighting surveys at 161 of our 183 long-term monitoring sites between December 2020 and May 2021 (Fig 1). The remaining 22 sites were inaccessible because of limited road access. We conducted spotlighting surveys at least one hour after sunset and during the period between 9 pm and 12 am. This allowed animals enough time to emerge from their dens [31] and therefore to be detected while outside tree hollows. Surveys were conducted along a 300 metre transect on the road immediately adjacent to the long-term monitoring site. Both sides of the road were surveyed at a pace of 10 minutes per 100 metres (not including recording time) as per the Victorian Government (Department of Environment, Land, Water, and Planning) survey guidelines (DELWP 2020 [32]). We used Olight Javelot Pro spotlights

which have a maximum throw of 1,080 metres and a maximum brightness of 2,100 lumens, allowing us to detect animal eye shine well into the forest from the transect. We surveyed for all arboreal marsupials and recorded species, abundance, distance from transect, bearing from sighting location, and location in forest vegetation (canopy, middle, lower branches, ground). We did not conduct spotlighting surveys during periods when it was raining, foggy, or windy.

Ethics approval for the field surveys of the Southern Greater Glider were provided by The Australian National University Animal Ethics Committee (protocols A2018/45 and A2021/15).

### Covariates for use in statistical analyses

We considered a suite of factors expected to influence the presence and abundance of Southern Greater Glider. Our potential explanatory variables included multiple environmental covariates derived from a LiDAR dataset from the Central Highlands of Victoria: slope, aspect, and elevation [33]. We present the range of values for these measures across our 161 field sites in S1 Table.

We estimated values for three climatic measures (corresponding to extreme conditions) to which a heat-sensitive species such as the South Greater Glider [27] might be expected to respond [13]. The first was the number of days between the start of 2015 and the end of 2019 that the daily maximum temperature was above 35˚C. Such extreme temperatures will typically correspond to day-time temperatures during which gliders may be at risk of heat stress whilst in their den trees. Our second measure was the number of days between 2015 and 2019 when minimum temperatures remained above 20˚C. The Southern Greater Glider is active only at night, and night-time temperature above 20˚C may result in decreased food intake to limit diet induced thermogenesis on these hot nights (see [13]). Other studies have demonstrated that night-time temperatures above 20˚C are associated with Southern Greater Glider population declines [28]. The climate values were interpolated data from climate surfaces generated from the Central Highlands region across Victoria and provided by the Commonwealth Scientific and Industrial Research Organisation (CSIRO) for daily minimum [34] and maximum [35] surface temperatures using combined MODIS LTS and local topography data [36,37].

Values for the number of hollow-bearing trees at a site was derived from an on-the-ground count (see [14]). We defined a hollow-bearing tree as any live/dead tree > 80 cm DBH and containing obvious hollows as determined by scanning from the ground using binoculars. Notably, some areas of young forest supported hollow-bearing trees, likely because the forest was old growth at the time it was burnt [23]. Finally, we estimated a crude score (low, medium, and high) of vegetation density in the understorey and overstorey in an effort to assess if effects on the ability to detect animals with a spotlight.

### Statistical analysis

We focused on a generalized linear mixed model approach, in part because logistical constraints meant that only one spotlighting survey per transect could be completed, and this made it impossible to implement distance sampling and detection/occupancy methods. We fit a hurdle Poisson model [38] to the numbers of the Southern Greater Glider recorded in each spotlighting transect. Our initial analyses indicated that the Southern Greater Glider was absent from sites dominated by Shining Gum and extremely rare in sites dominated by Alpine Ash forest. We therefore elected to restrict all subsequent statistical analyses to our 123 sites that were dominated by Mountain Ash forests. On this basis, let $y_i$ represent the number of the Southern Greater Glider recorded on site $i$ ($i = 1, . . ., 123$). The hurdle Poisson model consisted of two processes or components: **(1)** hurdle component in which we modelled the factors

associated with the probability of a zero count at site $i$ and **(2)** conditional abundance component in which we modelled the factors associated with the number of animals present. The hurdle component is modelled with a logistic regression model and the conditional abundance component is modelled with a zero-truncated Poisson regression model, since zeros are excluded from the first modelling step.

Let $p(\mathbf{x}_i)$ model the probability of a zero count on the *ith* site which depends on the vector of covariates $\mathbf{x}_i$, specifically the model is the following:

$$logit(p(\mathbf{x}_i)) = log\left(\frac{p(x_i)}{1 - p(\mathbf{x}_i)}\right) = \mathbf{x}_i'\mathbf{\beta}$$

where logit is the logistic transformation, $\mathbf{\beta}$ is the vector of regression parameters for the hurdle component of the model. Let $\lambda(\mathbf{z}_i)$ be the mean of the zero-truncated Poisson distribution with covariates $\mathbf{z}_i$, which we use to model the conditional abundance component, thus our model was the following:

$$log(\lambda(\mathbf{z}_i)) = \mathbf{z}_i'\mathbf{\gamma}$$

where $\mathbf{\gamma}$ is the vector of regression parameters for the conditional abundance portion of the model. The unconditional mean of $y_i$, which we denote by E($y_i$), can be expressed as a function of both $\mathbf{x}_i$ and $\mathbf{z}_i$, as follows:

$$\text{E}(y_i) = \lambda(\mathbf{x}_i, z_i) = \frac{\lambda(\mathbf{z}_i)(1 - p(\mathbf{x}_i))}{1 - \exp(-\lambda(\mathbf{z}_i))} \qquad \text{Eq 1}$$

where the denominator is the probability of a zero from the non-truncated Poisson distribution.

Welsh, 1996 and Welsh et. al., 1996 [38,39] showed that the likelihood for this hurdle Poisson model factors into a product of the likelihood for the hurdle component and the conditional abundance component allowing estimation to proceed separately. The approach allows for model selection to be performed independently on the two components. We considered the same set of covariates for each model component and performed model selection using the leave one out information criteria (LOOIC) [40,41] separately for each component. LOOIC can be seen as generalization of AIC to Bayesian models and has a similar interpretation, that is, models with lower LOOIC are deemed to fit the data better. For both stages of the model selection, we chose the most parsimonious model, defined as the model with the fewest parameters within two LOOIC units of the model with the lowest LOOIC. We considered the following 32 models for each of the two model components. There was a high degree of correlation among several variables including elevation, the number of days when the maximum temperature was greater than 35˚C, and the number of days the minimum temperature was greater than 20˚C (see S1 Table). Given this, we did not include more than one of these variables at a time in each of our 32 models. Our 32 models can be broken down into four sets of eight models each: **(A)** all possible combinations of the remaining three variables: vegetation density (low, medium, high), forest age, and the number of hollow-bearing trees on the site; **(B)** models from set (A) + elevation; **(C)** models from set (A) + number of days the maximum temperature is greater than 35˚C. And, **(D)** models from set (A) + number of days the minimum temperature is greater than 20˚C. We chose the most parsimonious model, that is, the simplest model within two LOOIC units of the best fitting model.

We fit models using a Bayesian approach via the brms package [42,43] in R [44] version 4.0.5. We used Student-t priors with seven degrees of freedom with zero location and scale of 2.5 for all regression parameters (after scaling the continuous variables) to avoid potential

problems with complete/partial separation (see https://github.com/stan-dev/stan/wiki/Prior-Choice-Recommendations). We ran 2000 iterations of four Markov chains with a burn in of 1000, leaving 4000 samples for posterior inference, which was assessed by the Gelman Rubin $\hat{R}$ statistic [45,46]. All parameters had $\hat{R}$'s $< 1.01$ indicating adequate mixing, which was confirmed by examining trace plots of all model parameters. We present posterior medians and 95% credible intervals for model parameters and for the combined effects of covariates.

Prior to constructing models for the presence and abundance of the Southern Greater Glider in our 123 Mountain Ash sites, we quantified correlations among potential explanatory variables, particularly elevation and our two temperature measures (see S2 Table). As expected, we found high negative correlations between elevation and the number of days between 2015 and 2019 that the daily maximum temperature was above 35˚C (R = -0.857) and between elevation and the number of days between 2015 and 2019 when minimum temperatures exceeded 20˚C (R = -0.729) (see S2 Table).

## Results

### General findings

We detected eight species of arboreal marsupials (Table 1). The vast majority were rare including the Critically Endangered Leadbeater's Possum (Table 1). We confined our analyses to the Southern Greater Glider which was rare in Alpine Ash forests and absent from Shining Gum forests. We therefore restricted our statistical analyses to sites dominated by Mountain Ash forest, which comprised 123 of our 161 sites. We present descriptive information for potential explanatory variables for these sites in S2 Table.

### Statistical model for the Southern Greater Glider

We fit 32 models (see S2 Table). Model selection revealed the best fitting and most parsimonious model for the probability of not detecting the Southern Greater Glider included a negative effect of increasing numbers of hollow-bearing trees (Table 2). There also was a forest age effect, with the probability of not detecting the Southern Greater Glider approaching one in stands regenerating after the 2009 wildfires (and which were ~ 12 years old at the time of our spotlighting surveys) (Table 2) (Fig 2A).

**Table 1. Numbers of individuals and sites at which different species of arboreal marsupials were detected.** We have listed species in order of detection frequency.

| Common name | Latin name | All sites | | Mtn Ash Sites only | |
|---|---|---|---|---|---|
| | | Number of sites where detected | Number of animals recorded | Number of sites where detected | Number of animals recorded |
| Southern Greater Glider | *Petauroides volans* | 39 | 88 | 34 | 79 |
| Mountain Brushtail Possum | *Trichosurus cunninghami* | 41 | 68 | 31 | 51 |
| Common Ringtail Possum | *Pseudocheirus peregrinus* | 31 | 53 | 23 | 42 |
| Yellow-bellied Glider | *Petaurus australis* | 25 | 45 | 16 | 28 |
| Kreft's Glider* | *Petaurus notatus* | 18 | 20 | 15 | 16 |
| Leadbeater's Possum | *Gymnobelideus leadbeateri* | 10 | 17 | 9 | 16 |
| Feathertail Glider | *Acrobates pygmaeus* | 2 | 2 | 1 | 1 |
| Common Brushtail Possum | *Trichosurus vulpecula* | 1 | 1 | 0 | 0 |

*Formerly known as the Sugar Glider (*Petaurus breviceps*).

**Table 2. The best fitting hurdle model for the presence and conditional abundance of the Southern Greater Glider in Mountain Ash forests of the Central Highlands of Victoria.** The hurdle component models the probability of recording zero individuals of the Southern Greater Glider on a site and the conditional component models the number of the Southern Greater Glider recorded given the presence of the species on a site. mASL = metres above sea level.

| Model Component | Parameter | Posterior Median | Lower 95% CI | Upper 95% CI |
|---|---|---|---|---|
| Conditional Abundance | Intercept | 0.61 | 0.30 | 0.89 |
| | Elevation (mASL) | 0.37 | 0.08 | 0.66 |
| Hurdle Component | Intercept | 0.57 | 0.04 | 1.09 |
| | No. of hollow-bearing trees | -0.87 | -1.59 | -0.18 |
| | ForestAge: 1960–1990 | 0.67 | -0.67 | 2.14 |
| | ForestAge: 2009 | 10.35 | 2.73 | 26.69 |
| | ForestAge: Old Growth | 0.56 | -1.26 | 2.44 |

The best fitting and most parsimonious model for the conditional abundance of the Southern Greater Glider (i.e. abundance given presence in the hurdle component of the model) included one covariate–elevation (Tables 2 and S2). The species was more abundant on sites at higher elevations (Fig 2C). Key effects in the combined model for unconditional abundance are displayed in Fig 2D and 2F and they show: **(a)** greater abundance on higher elevation sites (Fig 2D), **(b)** greater abundance on sites where hollow-bearing trees are prevalent (Fig 2E), and **(c)** an almost complete absence of detections of animals in 12-year-old forest that regenerated after wildfires in 2009 (Fig 2F).

## Discussion

### Tree hollow abundance effects

The hurdle part of our model revealed a negative relationship between the probability of not detecting the Southern Greater Glider and increasing numbers of hollow-bearing trees (Fig 2A). Hence, the species was more likely to be recorded on sites with many hollow-bearing trees (e.g. > 20 trees per site). This result was expected as the Southern Greater Glider is a cavity-dependent species [21–24,47] and previous studies in ash-type eucalypt forests (based on stag watching rather than spotlighting surveys) have highlighted strong relationships between the species being recorded and the abundance of hollow-bearing trees [14,48,49].

### Forest age effects

We found a forest age effect in the hurdle part of our analysis, with the Southern Greater Glider not detected on sites that had been subject to a high-severity, stand-replacing burn in the 2009 wildfires and which were ~12 years old at the time of our surveys (Fig 2A). This result is broadly consistent with other work which suggests that the Southern Greater Glider is sensitive to the effects of wildfire [14]. There was limited difference in the probability that the Southern Greater Glider would be absent from surveyed sites of other ages (Fig 2A). Our data suggest that stands may need to be at least ~ 30 years old before they are suitable for recolonization by the Southern Greater Glider. However, this age class requirement is nuanced because of the way the age cohorts of sites were classified and may not reflect the availability of key elements of habitat suitability for the Southern Greater Glider. We assigned each site an age based on an assessment of the age of the dominant live trees present. For example, stands dating from the 1980s-1990s are dominated by trees that are 30–40 years old, but they will only likely support individuals of the Southern Greater Glider if there are much older trees present in the stand (which often exceed 200–400 years old)–a biological legacy effect (*sensu* [50]). These large old hollow-bearing trees are required for denning and nesting by the Southern Greater Glider (see [21]) and the species will be absent from younger aged forests where such

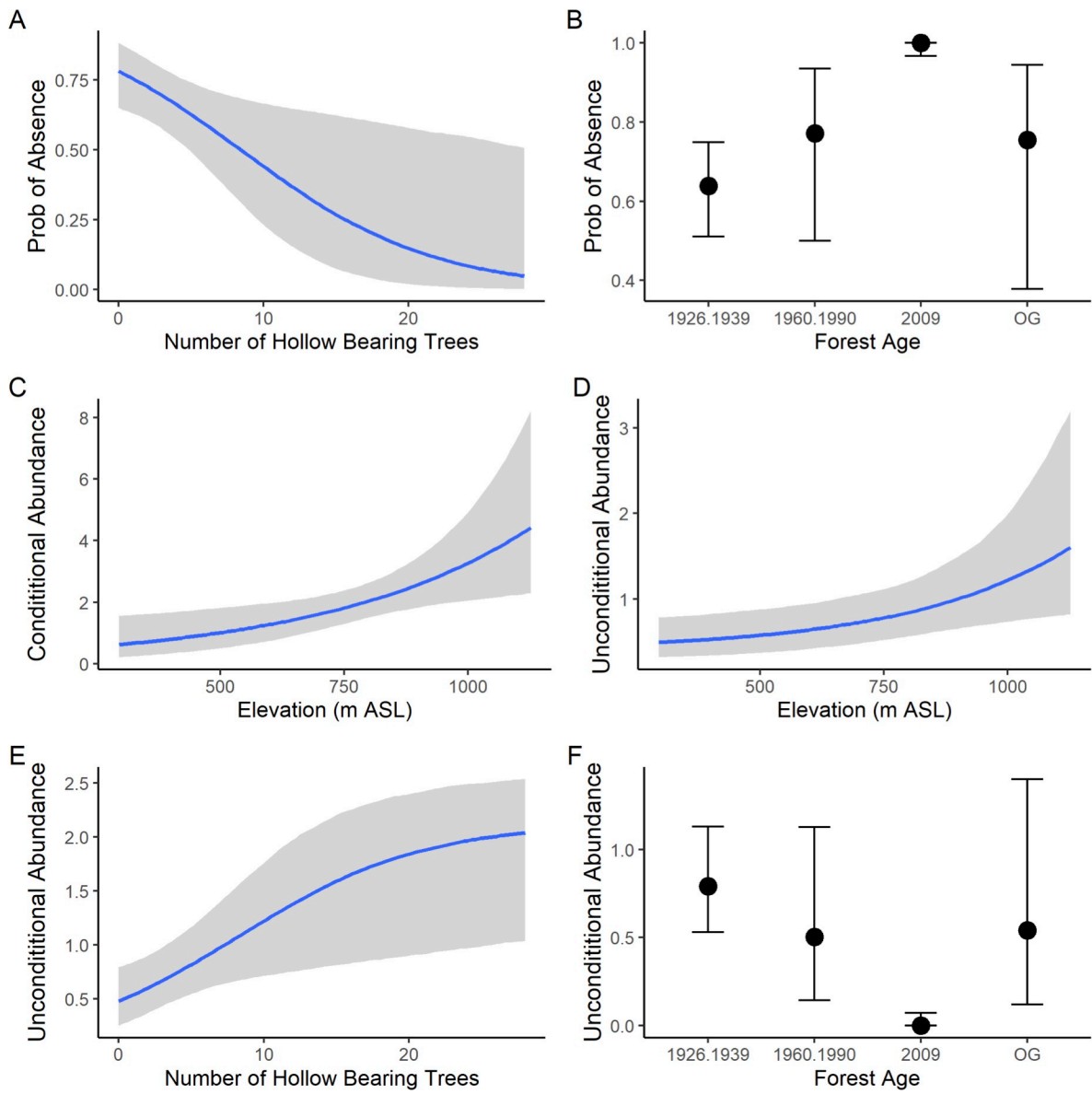

**Fig 2. The results of statistical modelling for the Southern Greater Glider.** The abundance of hollow-bearing trees (A) and forest age (B) in the hurdle portion of the model (i.e. the probability of the species being detected at a site). "Prob" = probability. The effect of Elevation at a site in the conditional abundance portion of the model (given the species is present) (C). Combined model (i.e. unconditional abundance) effects that include elevation (D), the abundance of hollow-bearing trees (E), and forest age (F) (OG = stands dating from before 1900, 1926–39 = stands dating from 1926–1939, 1960–90 = stands dating from 1960–1990 and 2009 = stands regenerating after the 2009 wildfires).

large old trees are rare or absent, as shown in this study (see Fig 2A). Past studies have shown that older forests support more such trees than younger stands [51,52]. However, analyses with interactions did not improve the model fit of either component.

## Elevation effects

We found that the conditional abundance of the Southern Greater Glider (that is, abundance given presence) increased with elevation (Fig 2B). Other studies have found that high elevation

sites can be important for the Southern Greater Glider [18,28]. We suggest that gliders are unlikely to be responding to elevation *per se*. Rather, elevation is likely a proxy for climate-related variables, especially since the species is known to be temperature sensitive [27]. Notably, we found that the temperature measures we considered were correlated with elevation, but elevation featured in the best fitting and most parsimonious model. This suggests that either elevation better predicted temperatures than the climate model or that factors beyond the specific climate variables we considered drove the elevation response. These may include rainfall (which is also correlated with elevation as well as temperature) and other temperature variables that were not considered.

Alpine Ash forests occur at higher elevations and experience cooler bioclimatic conditions than Mountain Ash [53,54]. While we identified a positive relationship between the conditional abundance of Southern Greater Glider and elevation in Mountain Ash forest, there was a paucity of the species in Alpine Ash forests (see also [55]). Therefore, forest type effects (with critical factors like food suitability varying between tree species; [19,56]) may be stronger than elevation effects at the upper end of the altitudinal range limits of some eucalypt tree species (i.e. the elevation-based replacement of Mountain Ash by Alpine Ash forest). However, elevation itself can also influence the nutritional quality of food, and it may not be the shift in vegetation species *per se*, but changes in foliar chemistry that are influenced by elevation (e.g., sodium availability) that could be responsible for the absence of greater gliders from the higher elevation sites that also happen to be dominated by a different eucalypt species [57].

The low number of records of the Southern Greater Glider in Alpine Ash forests, relative to Mountain Ash forests, also could be influenced by differences in the availability of hollow-bearing trees. Indeed, interspecific differences between the growth and development (and other biological processes e.g. fungal attack) of Alpine Ash and Mountain Ash trees may influence the development of hollows [58]. For instance, previous work has indicated that Mountain Ash forests may be more likely to support a higher number of hollow-bearing trees than Alpine Ash forests [52,59].

Additional research is required to determine whether the absence of greater gliders from Alpine Ash is due to species related differences in food quality, environmental factors that can influence food quality, or another reason unrelated to variations in the nutritional quality of those landscapes.

## Management implications

Our results have some important implications for Southern Greater Glider conservation and forest management. First, consistent with earlier studies (see [14,48,49]), there were strong relationships between the abundance of large old hollow-bearing trees and both the presence and the unconditional abundance of the Southern Greater Glider. However, the abundance of these kinds of trees has been declining rapidly in Mountain Ash forests, with numbers currently ~ 50% lower than they were two decades ago [51]. This means that populations of the Southern Greater Glider are likely to continue to decline in response to the increasing rarity of key shelter resources–as demonstrated in a recent time series study in the Central Highlands of Victoria [14]. Areas where such trees are most abundant–old growth forests–are themselves extremely rare, with just 1.16% of the Mountain Ash estate (with the equivalent figure of 0.47% for Alpine Ash) now old growth following recurrent wildfires and widespread clearcutting over the past 50–100 years [17]. We argue that far more stringent codes of forest practice are needed to better protect existing large, old hollow-bearing trees, such as with a buffer of unlogged forest. We base this recommendation on the fact that these trees are at elevated risk of collapse as the amount of logged forest in the landscape increases [60]. There also will be a

need for far greater efforts to protect advanced regrowth forest to eventually recruit more stands of old growth Mountain Ash forest. This is pertinent, as recent studies indicate that the probability of forests reaching older ages (~80 years) and developing adequate hollow-bearing trees (~180 years) is predicted to be as low as 0.03 (3% of fire intervals) under future fire regimes [61].

A second key implication of our study was that sites with the lowest conditional abundance of the Southern Greater Glider were at the lowest elevations of Mountain Ash forest. As outlined above, the ecological processes underpinning these patterns remain unclear. However, if they are physiologically based, and influenced by factors such as temperature, there may be altitudinal limits curtailing the extent of an upward movement in the species distribution in Mountain Ash forests. At high elevations where Mountain Ash forest is replaced by Alpine Ash, the Southern Greater Glider rarely occurs. This may be due to lower levels of abundance of hollow-bearing trees in Alpine Ash forests relative to Mountain Ash forests [62] and/or possible differences in the palatability of leaves between the two tree species for the Southern Greater Glider, and/or changes in the availability of key nutrients, like sodium, in response to elevation [57]. Targeted leaf sampling of Mountain Ash and Alpine Ash trees for nutritional quality analyses and feeding studies of captive individuals of the Southern Greater Glider will be required to determine if there are differences in leaf palatability between tree species and/or across elevational gradients.

Irrespective of the reasons for the elevation effects we identified, the findings of our study suggest that care will be needed to maintain intact parts of forest ecosystems with bioclimatic conditions that are suitable for occupancy by the Southern Greater Glider. Notably, recent studies indicate that the coolest and least variable microclimatic conditions in Mountain Ash forests occur in the oldest forests [63].

Populations of the Southern Greater Glider have been in marked decline in many parts of Australia over the past 20+ years, including in the Mountain Ash forests of the Central Highlands of Victoria [14]. The results of this study suggest that conservation efforts for the species in these forests may be best targeted at sites at higher elevations (which may act as climate refugia for the Southern Greater Glider) and in areas with numerous hollow-bearing trees.

## Supporting information

**S1 Table. Descriptive information for the covariates for all sites.** A) continuous variables, B) dominant tree species and C) Pearson correlations for the climate and elevation variables.
(DOCX)

**S2 Table. Descriptive information for the covariates for Mountain Ash sites.** (A) continuous variables, (B) forest age and (C) Pearson correlations for the climate and elevation variables.
(DOCX)

**S3 Table. Model selection results for the 24 models considered for the hurdle and conditional abundance components of the hurdle Poisson model.** The most parsimonious model is highlighted in bold (the simplest model within 2 ΔLOOIC units of the best fitting model).
(DOCX)

## Acknowledgments

Assistance with spotlighting surveys was provided by Dylan Lees. Tabitha Boyer and Jess Williams assisted with manuscript preparation. Comments from two anonymous referees and Ross Goldingay significantly improved earlier versions of the manuscript.

## Author Contributions

**Conceptualization:** David B. Lindenmayer, Lachlan McBurney, Wade Blanchard, Karen Marsh, Elle Bowd, Chris Taylor, Kara Youngentob.

**Data curation:** Lachlan McBurney, Darcy Watchorn.

**Formal analysis:** Wade Blanchard.

**Funding acquisition:** David B. Lindenmayer, Kara Youngentob.

**Investigation:** Elle Bowd, Kara Youngentob.

**Methodology:** David B. Lindenmayer, Wade Blanchard, Karen Marsh, Elle Bowd, Chris Taylor, Kara Youngentob.

**Project administration:** David B. Lindenmayer.

**Resources:** David B. Lindenmayer.

**Software:** Wade Blanchard.

**Supervision:** David B. Lindenmayer, Kara Youngentob.

**Validation:** Wade Blanchard, Kara Youngentob.

**Visualization:** Wade Blanchard, Chris Taylor.

**Writing – original draft:** David B. Lindenmayer, Karen Marsh, Elle Bowd, Chris Taylor, Kara Youngentob.

**Writing – review & editing:** David B. Lindenmayer, Lachlan McBurney, Wade Blanchard, Karen Marsh, Elle Bowd, Darcy Watchorn, Chris Taylor, Kara Youngentob.

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
