## [Decision Letter · Decision Letter 0]

8 Nov 2021

PONE-D-21-30842Elevation, disturbance, and forest type drive the occurrence of a specialist arboreal folivorePLOS ONE

Dear Dr. Lindenmayer,

Thank you for submitting your manuscript to PLOS ONE. After careful consideration, we feel that it has merit but does not fully meet PLOS ONE’s publication criteria as it currently stands. Therefore, we invite you to submit a revised version of the manuscript that addresses the points raised during the review process. In particular, both reviewers raised concerns regarding the decision to exclude some of the data from sites with lower detections and found the methods section to lack appropriate detail. Reviewer one provided useful comments regarding possible framing of the manuscript  that may help distinguish this study from previous research showing similar patterns. In addition, given that data was collected and described to account for covariates impacting detection it is not clear why the authors did not chose an analyses approach that could correct for differences in detectability. 

We look forward to receiving your revised manuscript.

Kind regards,

Angela Marie White, Ph.D.

Academic Editor

PLOS ONE

Journal Requirements:

“This work has been funded by the Victorian Government Department of Environment, Land, Water and Planning. Funding was recieved by DBL.”

“This work has been funded by the Victorian Government Department of Environment, Land, Water and Planning. Assistance with spotlighting surveys was provided by Dylan Lees. Tabitha Boyer and Jess Williams assisted with manuscript preparation.”

We note that you have provided funding information within the Acknowledgements Section. Please note that funding information should not appear in the Acknowledgments section or other areas of your manuscript. We will only publish funding information present in the Funding Statement section of the online submission form.

“This work has been funded by the Victorian Government Department of Environment, Land, Water and Planning. Funding was recieved by DBL.”

6. We note that Figure 1 in your submission contain map images which may be copyrighted. All PLOS content is published under the Creative Commons Attribution License (CC BY 4.0), which means that the manuscript, images, and Supporting Information files will be freely available online, and any third party is permitted to access, download, copy, distribute, and use these materials in any way, even commercially, with proper attribution. For these reasons, we cannot publish previously copyrighted maps or satellite images created using proprietary data, such as Google software (Google Maps, Street View, and Earth). For more information, see our copyright guidelines: http://journals.plos.org/plosone/s/licenses-and-copyright.

Reviewers' comments:

Reviewer's Responses to Questions

**Comments to the Author**

1. Is the manuscript technically sound, and do the data support the conclusions?

Reviewer #1: Partly

Reviewer #2: Partly

2. Has the statistical analysis been performed appropriately and rigorously? 

Reviewer #1: I Don't Know

Reviewer #2: Yes

3. Have the authors made all data underlying the findings in their manuscript fully available?

Reviewer #1: No

Reviewer #2: No

4. Is the manuscript presented in an intelligible fashion and written in standard English?

Reviewer #1: Yes

Reviewer #2: Yes

5. Review Comments to the Author

Reviewer #1: Overall:

The authors conducted a study of a threatened species facing several threats. Given the Southern Greater Glider’s association with higher elevations, climate change made add to these threats. As such, I think this is an important contribution, but would require some major revisions prior to publication.

General comments:

I think the manuscript could be greatly improved by more clearly framing the study around some specific objectives. Currently, it reads as if some long-term monitoring data were used to look for correlations. For example, they could more clearly establish early on in the introduction that Southern Greater Gliders are associated with high elevations, sensitive to temperature, and associated with older forests. As such, they are quite vulnerable to a warming global climate.

While the methods are mostly sound, some things need further justification, such as how vegetation density was assessed. In addition, more information about the statistical approach is required. Given that they mention that distances to animals were recorded, I would think a distance sampling framework that allows for the estimation of detection probability would be more appropriate.

In addition to these broader comments, more specific issues are listed below.

Abstract

My primary concern in this section is that I found the abstract to be a bit confusing, mostly regarding methods. First, I would suggest revising the first few sentences to clearly state the aim of the study, as opposed to “describe results.”

Line 26-28: Consider omitting the list of forest types to simply state the spotlighting surveys were conducted in southeastern Australia.

Line 28-30: Consider stating that the data to which the models were fit were comprised of counts of Southern Greater Glider along transects. Additionally, you might more clearly described how the models first estimate the probability of absence, and then if present, abundance is estimated. I realize this is described in more detail in the Methods, but it would be good to have this in the Abstract to give readers a better idea of what the study entailed. Further, some readers might be confused as to why further down within the Abstract you describe both absence and abundance, which are more commonly estimated with different model frameworks (e.g. occupancy models and distance sampling, respectively).

Line 39-40: This sentence seems tangential, and maybe does not need to be included in the Abstract.

Line 43-44: Was this an objective of the study? Perhaps this would be a more appropriate way to frame the study at the beginning of the Abstract?

Line 45-46: In the phrase, “…expect where they transition,” it is unclear to what “they” refers. Perhaps revise to something like, “…suggests that suitable forests at higher elevations will become increasingly important to the conservation of Southern Greater Glider.”

Introduction

Line 54-56: Are multiple spatial and temporal scales investigated here? If not, I would suggest omitting this sentence and combining the first two paragraphs into one.

Line 64-65: This sentence could be improved by revising to avoid passive voice.

Line 66-70: This sentence is quite difficult to read as written. Consider splitting into two or using punctuation to distinguish clauses. In addition, the subject of the first clause is “the conservation status,” yet “it” in the second clause appears to refer to the Southern Greater Glider. Avoiding these sorts of vague pronouns will improve clarity.

Line 70-75: These sentences could be improved by avoiding passive voice.

Line 77-78: This sentence is a bit awkward as written. Consider revising to something like, “Given these numerous threats, a better understanding of the factors that influence the abundance and distribution of Southern Greater Glider will become increasingly important to their conservation.”

Line 79-81: This objective seems inconsistent with the Abstract, which lists several other forest types. Also, there is no mention of abundance here, yet abundance is a focus of the Results section (e.g., lines 239-245). Given the central role that the objectives take on in a manuscript, I believe it is critical that these inconsistencies are resolved, both here and throughout the manuscript.

Methods

Line 128-129: Perhaps a more general topic sentence would be more appropriate for this paragraph. For example, “we considered a suite of factors that we expected to influence occurrence/abundance of Southern Greater Glider.” Also, please be consistent with Southern Greater Glider vs. Greater Glider. Is there ecological justification for incorporating these topographic metrics? If so, consider including that justification here.

Line 133-137: Can you provide a citation to support this claim?

Line 138-139: Consider a more concise language, e.g., “…the number of days when minimum temperature was above 20.” What was the spatial resolution in these climate data? Given the small study area, is there reason to believe that these metric would vary for any reason other than elevation? Given the high degree of correlation mentioned in the results, and no other mention of these variable in the results, I would suggest omitting this from the study, as it provides little information.

Line 151-153: Consider combining this with the discussion of tree age on lines 104-110.

Line 154-156: A paragraph is typically comprised of at least 3 sentences. Could this be merged with an existing paragraph? Perhaps the paragraph describing the environmental covariates of slope, aspect, and elevation?

Line 158-160: A paragraph is typically comprised of at least 3 sentences. Could this be moved to the section describing the spotlight surveys? Furthermore, please provide some description of how these categories were defined, and how consistent categorization between observers was ensured.

Line 162-166: Given the potential confusion between “mountain ash” and “Mountain Ash” I would suggest stating at the very beginning of the methods that the sample size is 123. You can then mention that this was part of a broader long-term sampling project, but including that information as written and the omission of the 22 sites due to access all just leads to unnecessary information that only serves to detract from the overall clarity of the manuscript.

Line 166-167: It is unclear why yi is defined when it is not referenced anywhere else in the manuscript.

Line 168-173: Please provide a sentence or two to justify why this modelling framework was selected, rather than more widely-used modelling frameworks, such as occupancy or distance sampling.

Line 191-205: Some justification why this less widely-used model comparison criterion was used is needed. In addition, some description of how the LOOIC is used to compare models (e.g., lower = better fit? More parsimonious? Both?

Line 206-207: Please revise to clarify that the models were fit to the data analyzed in a Bayesian setting.

Line 208: Please provide a citation for these priors.

Line 209: Chains are typically much longer. Did you inspect trace plots for convergence? Were any other model diagnostics assessed?

Results

Line 215-217: This paragraph, as well as Table 1, does not appear to be relevant to the manuscript and should be omitted.

Line 218-221: This information should be moved to the Methods.

Line 226-228: This sentence should be moved to the Methods section.

Line 242-245: It is unclear why the top-ranked model of abundance includes only elevation, yet the relationships between abundance and each of the number of hollow-bearing trees and forest age are also discussed. The “combined” model should be discussed more clearly in the Methods section. Further, there appears to be a great deal of uncertainty around the effects of the number of hollow bearing trees and forest age. As I commented elsewhere, it may be more appropriate to condense these variable into fewer categories.

Discussion

Line 272-273: The claim that they are more likely to be observed on sites with large numbers of hollow bearing trees in problematic. First, the distribution of the data is poor, such that there is a great deal of uncertainty around this relationship, particularly at sites where there were more hollow-bearing trees. Second, what is meant by “large number?” 5? 10? Third, I would avoid interchanging observed/occurred. Since you did not estimate the probability an animal was observed, given present, I would avoid that term.

Line 277-294: Based on the text of this paragraph, I would encourage the authors to investigate an interaction between forest age and the number of hollow-bearing trees.

Line 315-319: It seems like the authors would be able to investigate this relationship with their data (i.e. do Moutnain Ash forests have more hollow-bearing trees than Alpine Ash?). I would encourage them to think more carefully about why Greater Gliders to no occur in Alpine Ash forests, despite being suitable elevation.

Line 344-347: This sentence is awkward and should be revised.

Line 349: It is unclear to what “they” refers. Please avoid use of these vague pronouns.

Figures

Figure 1. I would remove the sites dominated by other tree species as they do not appear to be relevant to this study that focused on the Mountain Ash sites only. Additionally, it may be helpful to readers outside Australia to have an inset map showing the geographical context. Finally, there is nothing in the legend or caption about what the border represents. (presumably some conservation area?)

Figure 2. From this graphic and the table in the supplement, it seems that the data for the Nu. Of hollow bearing trees is not well distributed. Perhaps converting this to a binary variable would be more appropriate? (e.g. hollow bearing trees present or not). Also, two of the panels lack labels (e.g. Figure 2d and Figure 2e)

Reviewer #2: Review of PONE-D-21-30842

Elevation, disturbance, and forest type drive the occurrence of a specialist arboreal folivore

I enjoyed this manuscript, which investigated habitat use in a species of conservation concern. As such, the question addressed is important, and the findings of importance. The manuscript is well-written and the methods appear sound. I really liked the explicit management implications section.

I had the following minor comments that may help clarify some aspects of the manuscript.

1. I was a little confused about the survey procedure. The surveys were conducted at least one hour after sunset (9pm-12am), but during some of the months (Dec/Jan), it would not have been dark by 9pm, or possibly just getting dark. Were the surveys started later by season? Do the surveys last from 9-12 or were just conducted sometime between 9 and 12? Does glider activity change throughout that 3 hour period? In at least some nocturnal species, activity varies throughout the night. I’m looking here for clarity of what was conducted, and some reassurance that time of night would not have effected glider detectability.

2. All surveys were conducted along the road. Is there any evidence of either attraction to or avoidance of roads? Is habitat use along the road likely to be similar to areas away from roads?

3. Methods/Results – I was confused why only 123 of the 161 sites were included, and particularly why this information was introduced in the results. I think this should in the methods when you talked about the 161 sites, since this is not really a result. The results presented here (Table 1) could also be used to justify the inclusion of mountain ash sites only if presented in the methods. Alternatively, I would consider including an analysis in all habitat types since these may confirm your described patterns, or suggest other factors in other habitat types. At the moment, 5 sites with 9 animals (~10% of both sites and animals) have been excluded from the analysis – but may have valuable information for future conservation. The latter approach is my preference – include more information on the non-mountain ash sites – since your main aim is to describe factors associated with occurrence in ash forests.

4. Discussion – first paragraph – I would have liked some consideration of why the species might be rare in Alpine Ash and absent in Shining Gum. The may be absent/rare because this is less preferred habitat, or instead due to other contributing factors that could shed light on threatening processes.

6. PLOS authors have the option to publish the peer review history of their article (what does this mean?). If published, this will include your full peer review and any attached files.

Reviewer #1: **Yes: **Paul Taillie

Reviewer #2: No

---

## [Author Response · Author response to Decision Letter 0]

12 Dec 2021

Re: Revisions to PONE-D-21-30842 Elevation, disturbance, and forest type drive the occurrence of a specialist arboreal folivore

Dear Dr Angela Marie White,

 Thankyou for your correspondence of 9th November about our paper PONE-D-21-30842 Elevation, disturbance, and forest type drive the occurrence of a specialist arboreal folivore. We were asked to make revisions to the paper and we have now carefully and thoroughly revised the manuscript, addressing all of the comments of Reviewer #1 and Reviewer #2. 

 We have made a large number of changes to the manuscript as requested and these have helped greatly strengthen the paper. Indeed, we are most grateful for the insightful suggestions from the reviewers on ways to improve the manuscript. 

More specifically, we have:

• Added considerable further background information to the Methods, Results and Discussion section as requested. 

• Much better explained some of the key field and statistical methods that were applied. 

• Clarified many additional points as indicated in the comments from Reviewer #1 and Reviewer #2. 

Our more detailed responses to the comments from the Reviewers are set out in the remainder of this letter. We believe that these additional modifications have helped further strengthen the paper. 

We trust that this letter and our further revised manuscript will be received favourably and look forward to hearing from you in the near future. 

Yours sincerely,

David Lindenmayer

(On behalf of all authors)

 

PONE-D-21-30842

Elevation, disturbance, and forest type drive the occurrence of a specialist arboreal folivore

Journal Requirements:

Response from authors

We have carefully followed the guidelines for the resubmission of our extensively revised article. 

Journal Requirements:

“This work has been funded by the Victorian Government Department of Environment, Land, Water and Planning. Funding was recieved by DBL.”

Journal Requirements:

“This work has been funded by the Victorian Government Department of Environment, Land, Water and Planning. Assistance with spotlighting surveys was provided by Dylan Lees. Tabitha Boyer and Jess Williams assisted with manuscript preparation.”

We note that you have provided funding information within the Acknowledgements Section. Please note that funding information should not appear in the Acknowledgments section or other areas of your manuscript. We will only publish funding information present in the Funding Statement section of the online submission form.

“This work has been funded by the Victorian Government Department of Environment, Land, Water and Planning. Funding was recieved by DBL.”

Response from authors

We have removed the text on funding from the Acknowledgments section as requested and will submit this information as part of the Online statement. 

Journal Requirements:

Response from authors

We wish to keep our data availability statement as previously outlined, and will provide a link to where the data can be accessed. 

Journal Requirements:

Response from authors

We have now provided an ethics committee statement as requested. 

Journal Requirements:

6. We note that Figure 1 in your submission contain map images which may be copyrighted. All PLOS content is published under the Creative Commons Attribution License (CC BY 4.0), which means that the manuscript, images, and Supporting Information files will be freely available online, and any third party is permitted to access, download, copy, distribute, and use these materials in any way, even commercially, with proper attribution. For these reasons, we cannot publish previously copyrighted maps or satellite images created using proprietary data, such as Google software (Google Maps, Street View, and Earth). For more information, see our copyright guidelines: http://journals.plos.org/plosone/s/licenses-and-copyright.

Response from authors

We have created our own figures for this article and there are no copyright issues. 

 

COMMENTS FROM REVIEWER #1

The authors conducted a study of a threatened species facing several threats. Given the Southern Greater Glider’s association with higher elevations, climate change made add to these threats. As such, I think this is an important contribution, but would require some major revisions prior to publication. 

Response from authors

We thank Reviewer #1 for their positive comments on our paper. However, we acknowledge that they have made a substantial number of suggestions and we have now carefully revised the manuscript on the basis of the comments they have made. 

Comments from Reviewer #1

General comments: 

I think the manuscript could be greatly improved by more clearly framing the study around some specific objectives. Currently, it reads as if some long-term monitoring data were used to look for correlations. For example, they could more clearly establish early on in the introduction that Southern Greater Gliders are associated with high elevations, sensitive to temperature, and associated with older forests. As such, they are quite vulnerable to a warming global climate. 

Response from authors

This was a good suggestion from Reviewer #1 and we have now revised the manuscript to help better frame our work and highlight that, for example, sensitivity to higher temperatures may underscore the value of higher elevation areas for the Greater Glider. 

Comments from Reviewer #1

While the methods are mostly sound, some things need further justification, such as how vegetation density was assessed. In addition, more information about the statistical approach is required. Given that they mention that distances to animals were recorded, I would think a distance sampling framework that allows for the estimation of detection probability would be more appropriate. 

In addition to these broader comments, more specific issues are listed below. 

Response from authors

We thank Reviewer #1 for these comments. We have added further details to the Methods section as requested. Reviewer #1 raised the issue of employing distance sampling or detection/occupancy analysis. We carefully considered this suggestion given that we have considerable experience in the use of both kinds of statistical approaches. In the case of detection/occupancy data, multiple visits would be required to account for imperfect detection, and then subsequent modelling could be used to quantify the factors influencing occupancy (given detection). 

Comments from Reviewer #1

Abstract 

My primary concern in this section is that I found the abstract to be a bit confusing, mostly regarding methods. First, I would suggest revising the first few sentences to clearly state the aim of the study, as opposed to “describe results.” 

Response from authors

This was a good suggestion from Reviewer #1 and we have now revised the opening sentences in the Abstract to make the intent of the study clearer and reduce the potential for confusion amongst readers. 

Comments from Reviewer #1

Line 26-28: Consider omitting the list of forest types to simply state the spotlighting surveys were conducted in southeastern Australia. 

Response from authors

We have edited the Abstract as suggested by Reviewer #1. 

Comments from Reviewer #1

Line 28-30: Consider stating that the data to which the models were fit were comprised of counts of Southern Greater Glider along transects. Additionally, you might more clearly described how the models first estimate the probability of absence, and then if present, abundance is estimated. I realize this is described in more detail in the Methods, but it would be good to have this in the Abstract to give readers a better idea of what the study entailed. Further, some readers might be confused as to why further down within the Abstract you describe both absence and abundance, which are more commonly estimated with different model frameworks (e.g. occupancy models and distance sampling, respectively). 

Response from authors

We have reframed the opening sentences of the Abstract as suggested by Reviewer #2, by better highlighting the aims of the work and then more clearly indicating that we constructed models of presence absence, and then if the species was present, models of abundance. 

Comments from Reviewer #1

Line 39-40: This sentence seems tangential, and maybe does not need to be included in the Abstract. 

Response from authors

We have removed this sentence from the Abstract as suggested by Reviewer #1. 

Comments from Reviewer #1

Line 43-44: Was this an objective of the study? Perhaps this would be a more appropriate way to frame the study at the beginning of the Abstract? 

Response from authors

We have removed the reference to recurrent wildfires in the Abstract and revisited this issue (as it influences stand age) later in the paper. This was a good suggestion from Reviewer #1 and it helped better focus the Abstract. 

Comments from Reviewer #1

Line 45-46: In the phrase, “…expect where they transition,” it is unclear to what “they” refers. Perhaps revise to something like, “…suggests that suitable forests at higher elevations will become increasingly important to the conservation of Southern Greater Glider.” 

Response from authors

“They”, in this case, refers to the transition from Mountain Ash to Alpine Ash at elevational boundaries between the two forest types. We have now made this clearer in the revised manuscript. The revised text now reads: 

The influence of elevation on conditional abundance suggests that areas at higher elevations will be increasingly important for the conservation of the species, except where Mountain Ash forest is replaced by different tree species that may be unsuitable for the Southern Greater Glider. 

Comments from Reviewer #1

Introduction 

Line 54-56: Are multiple spatial and temporal scales investigated here? If not, I would suggest omitting this sentence and combining the first two paragraphs into one. 

Response from authors

This was a good question from Reviewer #1 here. Multiple spatial scales are indeed having an important influence here – from the availability of individual tree hollows to broader climatic conditions. We have therefore elected to keep the term “multiple spatial temporal scales” in the opening paragraph. 

Comments from Reviewer #1

Line 64-65: This sentence could be improved by revising to avoid passive voice. 

Response from authors

We have rewritten this sentence in a more active voice as requested by Reviewer #1. 

Comments from Reviewer #1

Line 66-70: This sentence is quite difficult to read as written. Consider splitting into two or using punctuation to distinguish clauses. In addition, the subject of the first clause is “the conservation status,” yet “it” in the second clause appears to refer to the Southern Greater Glider. Avoiding these sorts of vague pronouns will improve clarity. 

Response from authors

We have reworked the text by using a more active voice, reducing the sentence length, and dispensing vague pronouns as recommended by Reviewer #1. 

Comments from Reviewer #1

Line 70-75: These sentences could be improved by avoiding passive voice. 

Response from authors

We have rewritten the text on Lines 70-75 so that active voice is now used more prominently. 

Comments from Reviewer #1

Line 77-78: This sentence is a bit awkward as written. Consider revising to something like, “Given these numerous threats, a better understanding of the factors that influence the abundance and distribution of Southern Greater Glider will become increasingly important to their conservation.” 

Response from authors

We have reworked this text as suggested by Reviewer #1.

Comments from Reviewer #1

Line 79-81: This objective seems inconsistent with the Abstract, which lists several other forest types. Also, there is no mention of abundance here, yet abundance is a focus of the Results section (e.g., lines 239-245). Given the central role that the objectives take on in a manuscript, I believe it is critical that these inconsistencies are resolved, both here and throughout the manuscript. 

Response from authors

We agree that using the terms presence and abundance should have been more consistently applied throughout the text. We have carefully revised the manuscript to make sure that the appropriate terms are employed. 

Comments from Reviewer #1

Methods 

Line 128-129: Perhaps a more general topic sentence would be more appropriate for this paragraph. For example, “we considered a suite of factors that we expected to influence occurrence/abundance of Southern Greater Glider.” Also, please be consistent with Southern Greater Glider vs. Greater Glider. Is there ecological justification for incorporating these topographic metrics? If so, consider including that justification here. 

Response from authors

This was a good suggestion from Reviewer #1 and we have modified the text to create a better topic sentence, and provide better ecological motivation for the kinds of covariates that were tested. We have also now consistently used the common name Southern Greater Glider throughout the revised manuscript. 

Comments from Reviewer #1

Line 133-137: Can you provide a citation to support this claim? 

Response from authors

As requested, we have now included further citations concerning the heat sensitivity of the Southern Greater Glider. 

Comments from Reviewer #1

Line 138-139: Consider a more concise language, e.g., “…the number of days when minimum temperature was above 20.” What was the spatial resolution in these climate data? Given the small study area, is there reason to believe that these metric would vary for any reason other than elevation? Given the high degree of correlation mentioned in the results, and no other mention of these variable in the results, I would suggest omitting this from the study, as it provides little information. 

Response from authors

We have simplified the language as suggested by Reviewer #1. We considered the suggestion from Reviewer #1 that commentary on this variable should be omitted, but it was tested in our statistical modelling and we therefore believe that it would be inappropriate to simply delete it. We note that we have been conducted detailed studies using site-level temperature loggers and there are indeed important temperature variations that need to be considered. 

Comments from Reviewer #1

Line 151-153: Consider combining this with the discussion of tree age on lines 104-110. 

Response from authors

This was a good suggestion from Reviewer #1 and we have moved the text to earlier in the Methods as recommended. 

Comments from Reviewer #1

Line 154-156: A paragraph is typically comprised of at least 3 sentences. Could this be merged with an existing paragraph? Perhaps the paragraph describing the environmental covariates of slope, aspect, and elevation? 

Response from authors

This was a good suggestion from Reviewer #1 and we have combined the short paragraphs as suggested. 

Comments from Reviewer #1

Line 158-160: A paragraph is typically comprised of at least 3 sentences. Could this be moved to the section describing the spotlight surveys? Furthermore, please provide some description of how these categories were defined, and how consistent categorization between observers was ensured. 

Response from authors

We have now added several sentences to the revised manuscript to clarify the definition of categories as requested by Reviewer #1. 

Comments from Reviewer #1

Line 162-166: Given the potential confusion between “mountain ash” and “Mountain Ash” I would suggest stating at the very beginning of the methods that the sample size is 123. You can then mention that this was part of a broader long-term sampling project, but including that information as written and the omission of the 22 sites due to access all just leads to unnecessary information that only serves to detract from the overall clarity of the manuscript. 

Response from authors

We thank Reviewer #1 for this comment. We surveyed 161 sites, but those dominated by Alpine Ash and Shining Gum supported almost no animals. This is an important result – and we do strongly consider that it is important for conservation to report this result – as we discuss further in the revised Discussion section of the manuscript. However, given the paucity of records in these two forest types, we were unable to include data from Alpine Ash and Shining Gum in the statistical analysis. The information about the 38 non Mountain Ash forests is critical to report – even if it is largely a null result with few animals. This then leads to the initial parts of the revised Discussion section where we consider why the Southern Greater Glider is so uncommon in Alpine Ash and Shining Gum forest. 

Given these comments from Reviewer #1, and to avoid confusion in the text, we now make it clearer where we are dealing with Mountain Ash forest, Alpine Ash forest, and Shining Gum forest. 

Comments from Reviewer #1

Line 166-167: It is unclear why yi is defined when it is not referenced anywhere else in the manuscript. 

Response from authors

We have now revised the manuscript and referred to yi on lines 199-201 in the revised paper to make the connection between the counts and the mean of the hurdle Poisson model. 

Comments from Reviewer #1

Line 168-173: Please provide a sentence or two to justify why this modelling framework was selected, rather than more widely-used modelling frameworks, such as occupancy or distance sampling. 

Response from authors

We focused on a generalized linear mixed model approach for a few reasons, though, we do acknowledge detection issues are problematic. First, due to time and other constraints, we completed only one spotlighting survey per transect and this makes it impossible to implement distance/sampling and detection occupancy. Distance sampling has some issues associated with it as pointed out by Barry and Welsh (2001). Notably, the confounding between the detection function and the spatial distribution of the animal. They state that “We cannot tell on the basis of the observed distance data whether, when we observe only a few objects, it is because there are only a few objects to observe or because detection is poor”. 

Given the comments of Reviewer #1, we have added a new sentence at the start of the Statistical Analysis section outlined why we employed a generalized linear mixed model approach. 

Distance Sampling Methodology Author(s): Simon C. Barry and A. H. Welsh Source: Journal of the Royal Statistical Society. Series B (Statistical Methodology) , 2001, Vol. 63, No. 1 (2001), pp. 31-53 Published by: Wiley for the Royal Statistical Society Stable URL: https://www.jstor.org/stable/2680632

Comments from Reviewer #1

Line 191-205: Some justification why this less widely-used model comparison criterion was used is needed. In addition, some description of how the LOOIC is used to compare models (e.g., lower = better fit? More parsimonious? Both? 

Response from authors

This was a useful suggestion from Reviewer #1 and we have added some further text to the Statistical Analyses section about the use of LOOIC. The revised text now states:

LOOIC can be seen as generalization of AIC to Bayesian models and has a similar interpretation, that is, models with lower LOOIC are deemed to fit the data better. For both stages of the model selection, we choose the most parsimonious model, defined as the model with the fewest parameters within two LOOIC units of the model with the lowest LOOIC. 

Comments from Reviewer #1

Line 206-207: Please revise to clarify that the models were fit to the data analyzed in a Bayesian setting. 

Response from authors

The text in this section now reads:

We fit models using a Bayesian approach via the brms package [36, 37] in R [38] version 4.0.5.

Comments from Reviewer #1

Line 208: Please provide a citation for these priors. 

Response from authors

As requested by Reviewer #1, we have provided a citation for priors (= https://github.com/stan-dev/stan/wiki/Prior-Choice-Recommendations). 

Comments from Reviewer #1

Line 209: Chains are typically much longer. Did you inspect trace plots for convergence? Were any other model diagnostics assessed? 

Response from authors

The brms package uses the stan package to do the Markov chain Monte Carlo which uses Hamiltonian Monte Carlo (HMC) instead of traditional Gibbs/Metropolis Hastings samplers. HMC samplers require fewer samples than more traditional MCMC methods by reducing the correlation between samples while maintaining high acceptance probabilities Radford Neal (2001). 

Radford Neal (2011). MCMC using Hamiltonian Dynamics, in Handbook of Markov Chain Monte Carlo, edited by Steve Brooks, Andrew Gelman, Gailin L. Jones Xiao-Li Meng, CRC Press. 

We have not added this extra detail to the manuscript, but we are happy to do so if the Editor and Reviewer deem it appropriate to do so. Note that toward the end of this paragraph we have added some extra text to indicate that there was adequate mixing as confirmed by examining trace plots for all model parameters (see Line 240). 

Comments from Reviewer #1

Results 

Line 215-217: This paragraph, as well as Table 1, does not appear to be relevant to the manuscript and should be omitted. 

Response from authors

There is enormous interest from the Victorian and Australian Governments about survey results of observations of arboreal marsupials in Mountain Ash forests. Summary data on observed numbers of animals are therefore most important to present, including for Critically Endangered animals like Leadbeater’s Possum. We therefore would like to retain this information showing the actual data which underpinned our detailed statistical analyses for the Southern Greater Glider. 

Comments from Reviewer #1

Line 218-221: This information should be moved to the Methods. 

Response from authors

We have added some limited additional descriptive information about the paucity of animals in Alpine Ash and Shining Gum forests because it is a key result. We then highlight how we restricted our detailed statistical analyses to sites dominated by Mountain Ash forest. 

Comments from Reviewer #1

Line 226-228: This sentence should be moved to the Methods section. 

Response from authors

As requested by Reviewer #1, we have now moved this information on correlations between covariates to the Methods section. 

Comments from Reviewer #1

Line 242-245: It is unclear why the top-ranked model of abundance includes only elevation, yet the relationships between abundance and each of the number of hollow-bearing trees and forest age are also discussed. The “combined” model should be discussed more clearly in the Methods section. Further, there appears to be a great deal of uncertainty around the effects of the number of hollow bearing trees and forest age. As I commented elsewhere, it may be more appropriate to condense these variable into fewer categories. 

Response from authors

We believe that Reviewer #1 might have been confused about the way the statistical analysis was conducted and how our multiple-part hurdle modelling was implemented with a presence/absence only component, a conditional abundance (given presence), and then a combined analysis. This confusion was created in the original version of the manuscript by us not more clearly describing what statistical analyses were undertaken. We have now rectified these problems by adding an extra paragraph at the Statistical Analyses section. The added paragraph reads:

In the Results section below, we discuss the hurdle component of the model, that is, the factors that are associated with the detection of the Southern Greater Glider on a site and the conditional abundance component of the model, that is, the factors that are associated with the number of animals detected on the site given at least one was detected. We also present the combined or unconditional model, which includes factors from the hurdle component of the model and the conditional abundance component of the model as these factors are combined to give the unconditional mean number of animals detected at a site (see equation 1).

Comments from Reviewer #1

Discussion 

Line 272-273: The claim that they are more likely to be observed on sites with large numbers of hollow bearing trees in problematic. First, the distribution of the data is poor, such that there is a great deal of uncertainty around this relationship, particularly at sites where there were more hollow-bearing trees. Second, what is meant by “large number?” 5? 10? Third, I would avoid interchanging observed/occurred. Since you did not estimate the probability an animal was observed, given present, I would avoid that term. 

Response from authors

We have amended the text and exchanged the words observed and occurrence for “recorded” wherever possible. We have clarified the descriptor “large” number here to be sites with > 20 hollow-bearing trees per site. This is now made clear on Line 328 in our extensively revised manuscript. 

Comments from Reviewer #1

Line 277-294: Based on the text of this paragraph, I would encourage the authors to investigate an interaction between forest age and the number of hollow-bearing trees. 

Response from authors

This was an insightful suggestion from Reviewer #1. We trialled doing this and the added models with interactions did not improve the model fit of either component. Given the comments from Reviewer #1, we have added some text on the potential for interactions to influence counts of the Southern Greater Glider. The additional text is provided on Lines 350-355 and reads: 

We acknowledge that there may be an interaction between the number of hollow-bearing trees and stand age that could influence detections of the Southern Greater Glider. Indeed, past studies have shown that older forests support more such trees than younger stands {Lindenmayer, 2018 #21}. However, analyses with interactions did not improve the model fit of either component. 

Comments from Reviewer #1

Line 315-319: It seems like the authors would be able to investigate this relationship with their data (i.e. do Moutnain Ash forests have more hollow-bearing trees than Alpine Ash?). I would encourage them to think more carefully about why Greater Gliders to no occur in Alpine Ash forests, despite being suitable elevation. 

Response from authors

This was a good suggestion from Reviewer #1. Previous work based on a very large number of sites (> 520) and many 1000s of trees has clearly indicated that Alpine Ash forests support significantly fewer hollow-bearing trees relative to Mountain Ash forests. We have now made this point clearer in the revised text. 

Comments from Reviewer #1

Line 344-347: This sentence is awkward and should be revised. 

Response from authors

We have rewritten the text here as suggested by Reviewer #1. The revised text now reads: 

At high elevations where Mountain Ash forest is replaced by Alpine Ash, the Southern Greater Glider rarely occurs. This may be due to lower levels of abundance of hollow-bearing trees in Alpine Ash forests relative to Mountain Ash forests (Lindenmayer et al., 1993) and/or possible differences in the palatability of leaves between the two tree species for the Southern Greater Glider. Targeted leaf sampling of Mountain Ash and Alpine Ash trees for nutritional quality analyses and feeding studies of captive individuals of the Southern Greater Glider will be required to determine the existence of palatability differences.

Comments from Reviewer #1

Line 349: It is unclear to what “they” refers. Please avoid use of these vague pronouns. 

Response from authors

“They” here refers to the findings of our study. We have revised the text to make this clearer to readers. 

Comments from Reviewer #1

Figures 

Figure 1. I would remove the sites dominated by other tree species as they do not appear to be relevant to this study that focused on the Mountain Ash sites only. Additionally, it may be helpful to readers outside Australia to have an inset map showing the geographical context. Finally, there is nothing in the legend or caption about what the border represents. (presumably some conservation area?)

Response from authors

We have revised Figure 1 to include an inset of the location in Australia where the study took place. The boundary represents the limit of the region known as the Central Highlands of Victoria. Please note that we carefully considered the suggestion that the survey points dominated by forests other than Mountain Ash be dropped from the figure. However, we would like to retain these points as they were surveyed for the Southern Greater Glider and the paucity of animals from these places is actually a very important result – and such rarity meant we were unable to conduct formal analyses of those forest types. 

Comments from Reviewer #1

Figure 2. From this graphic and the table in the supplement, it seems that the data for the Nu. Of hollow bearing trees is not well distributed. Perhaps converting this to a binary variable would be more appropriate? (e.g. hollow bearing trees present or not). Also, two of the panels lack labels (e.g. Figure 2d and Figure 2e)

Response from authors

This was an interesting suggestion from Reviewer #1 and we carefully considered it in rechecking our analyses. At issue here is that the data on hollow-bearing trees are right-skewed – as recognized by Reviewer #1, with up to 29 such trees per site. Sites with multiple hollow-bearing trees are ecologically quite different from sites with just one tree – as indicated by the graph highlight the probability of absence and the number of hollow-bearing trees. Indeed, there are large changes in probability between 1 tree and 29 trees. On this basis, it is better to model the count of the number of hollow-bearing trees rather than create a binary measure (0,1) in this case. 

 

COMMENTS FROM REVIEWER #2

Comments from Reviewer #2

Reviewer #2: Review of PONE-D-21-30842 

Elevation, disturbance, and forest type drive the occurrence of a specialist arboreal folivore 

I enjoyed this manuscript, which investigated habitat use in a species of conservation concern. As such, the question addressed is important, and the findings of importance. The manuscript is well-written and the methods appear sound. I really liked the explicit management implications section. 

I had the following minor comments that may help clarify some aspects of the manuscript. 

Response from authors

We thank Reviewer #2 for their very positive comments. We have carefully considered their suggestions for minor revision and ensure that we have considered them fully as part of modifying the manuscript. 

Comments from Reviewer #2

1. I was a little confused about the survey procedure. The surveys were conducted at least one hour after sunset (9pm-12am), but during some of the months (Dec/Jan), it would not have been dark by 9pm, or possibly just getting dark. Were the surveys started later by season? Do the surveys last from 9-12 or were just conducted sometime between 9 and 12? Does glider activity change throughout that 3 hour period? In at least some nocturnal species, activity varies throughout the night. I’m looking here for clarity of what was conducted, and some reassurance that time of night would not have effected glider detectability. 

Response from authors

We conducted surveys starting an hour after sunset – to match with the time animals become active after dusk (see Lindenmayer et al. 1991). Given the timespan over several months when spotlighting surveys were completed, the timing to start was variable – but always between 9 and 12 pm. We have made this clearer in the revised manuscript. 

Comments from Reviewer #2

2. All surveys were conducted along the road. Is there any evidence of either attraction to or avoidance of roads? Is habitat use along the road likely to be similar to areas away from roads? 

Response from authors

Reviewer #2 is correct that all surveys were conducted along roads. However, there is currently no evidence in this ecosystem that habitat use is likely to be different close to, or far away from, roads. 

Comments from Reviewer #2

3. Methods/Results – I was confused why only 123 of the 161 sites were included, and particularly why this information was introduced in the results. I think this should in the methods when you talked about the 161 sites, since this is not really a result. The results presented here (Table 1) could also be used to justify the inclusion of mountain ash sites only if presented in the methods. Alternatively, I would consider including an analysis in all habitat types since these may confirm your described patterns, or suggest other factors in other habitat types. At the moment, 5 sites with 9 animals (~10% of both sites and animals) have been excluded from the analysis – but may have valuable information for future conservation. The latter approach is my preference – include more information on the non-mountain ash sites – since your main aim is to describe factors associated with occurrence in ash forests. 

Response from authors

We thank Reviewer #2 for this comment. We surveyed 161 sites, but the Southern Greater Glider was very rare in Alpine Ash forest and absent from Shining Gum forest. Given the paucity of records of the species in Alpine Ash forest and Shining Gum forest, we were unable to include data from these two forest types in the subsequent statistical analyses. We have further described the statistical rationale for doing this in the revised manuscript. We have also now further discussed the reasons why the species may be so rare in Alpine Ash and Shining Gum forests in the revised Discussion section – as per comment #4 below from Reviewer #2. 

Comments from Reviewer #2

4. Discussion – first paragraph – I would have liked some consideration of why the species might be rare in Alpine Ash and absent in Shining Gum. The may be absent/rare because this is less preferred habitat, or instead due to other contributing factors that could shed light on threatening processes.

Response from authors

This was a good suggestion from Reviewer #2 and we have added some further text to the revised paper, speculating as to the reasons why the Southern Greater Glider is so uncommon in Alpine Ash forests and remained undetected in Shining Gum forests. This extra material is set further down in the revised Discussion section. 

---

## [Decision Letter · Decision Letter 1]

23 Feb 2022

PONE-D-21-30842R1Elevation, disturbance, and forest type drive the occurrence of a specialist arboreal folivorePLOS ONE

Dear Dr. Lindenmayer,

Thank you for submitting your manuscript to PLOS ONE. After careful consideration, we feel that it has merit but does not fully meet PLOS ONE’s publication criteria as it currently stands. Therefore, we invite you to submit a revised version of the manuscript that addresses the points raised during the review process.

We look forward to receiving your revised manuscript.

Kind regards,

Angela Marie White, Ph.D.

Academic Editor

PLOS ONE

Journal Requirements:

Reviewers' comments:

Reviewer's Responses to Questions

**Comments to the Author**

1. If the authors have adequately addressed your comments raised in a previous round of review and you feel that this manuscript is now acceptable for publication, you may indicate that here to bypass the “Comments to the Author” section, enter your conflict of interest statement in the “Confidential to Editor” section, and submit your "Accept" recommendation.

Reviewer #2: All comments have been addressed

Reviewer #3: All comments have been addressed

2. Is the manuscript technically sound, and do the data support the conclusions?

Reviewer #2: Yes

Reviewer #3: Yes

3. Has the statistical analysis been performed appropriately and rigorously? 

Reviewer #2: Yes

Reviewer #3: Yes

4. Have the authors made all data underlying the findings in their manuscript fully available?

Reviewer #2: Yes

Reviewer #3: No

5. Is the manuscript presented in an intelligible fashion and written in standard English?

Reviewer #2: Yes

Reviewer #3: Yes

6. Review Comments to the Author

Reviewer #2: The authors have addressed all my concerns - I really appreciated the detailed response to reviewers, and the amendments have improved the manuscript

Reviewer #3: General comments

This is an interesting and worthwhile study. The authors have done a good job of revising the ms based on the reviews of previous referees. I think some further revision will enhance the quality and readability of this ms. Some new paragraphs seem redundant given other text that is presented. Elsewhere I found there is existing or new text that is not concise and should be deleted. In a few places I think some additional references could be cited to link with earlier research conducted on this species. These suggestions are documented below.

Detailed comments and corrections

The line numbers refer to the clean copy.

Abstract

L32-33. This sentence can be deleted ‘We excluded data collected in these forest types in further analysis.’

Introduction

L69. Add Kavanagh & Lambert (1990) Aust Wildl Res 17:285.

L70. Add Kavanagh & Wheeler (2004). p.413-25. In ‘The Biology of Australian Possums and Gliders’. Add Lindenmayer, et al. (2004). Wildlife Research 31, 569.

Methods

L116. Glider surveys were conducted during Dec 2020 to May 2021. Why are temperature data described for 2019-20 and yet the temperature data used in an analysis was from 2015-19?

L139. Table S1 describes 3 not 2 temperature variables.

Is it meaningful to go to 2 decimal places with all of the variables? In Table S2 as well.

Change ‘No’ to ‘No.’

L141 & 144. It would be helpful to know explicitly the period over which the number of days were estimated to avoid confusion (i.e. is it 1 Jan 2015 to 31 Dec 2019? i.e. 5 years not 4).

L148. ‘that remain’ needs some qualification.

L171. It would be helpful to the reader if you stated the benefits of modelling species’ absence rather than presence.

L199. It’s not clear whether there is justification to build models with 4 covariates.

L201-3. Awkward sentence structure. Perhaps put variables in brackets or simplify to be referred to as temperature variables.

L206-8. Do the levels in the variables need to be restated here?

L209-10. Given the high correlation between elevation and the 2 temperature variables it is not clear why all 3 should be retained. Two appear redundant. One temp variable certainly is.

L223-229. Delete. This seems to be covered more concisely at L201-3. Note at L223 ‘presence’ rather than ‘absence’ is used.

There was a high correlation between the 2 temperature variables so only one should be used.

L229-236. Delete. This section seems redundant given it precedes the results section in which one could concisely state what is presented, and leave any discussion of the models to the discussion.

Results

L240-46. This could be written more concisely. Some text seems repetitious of what is stated elsewhere (e.g. confining analysis to only the mountain ash forest). Describe the greater glider findings first.

L251. Delete ‘As outlined above,’

Table S3. It would help if the models were arranged from lowest to highest ∆LOOIC for one column.

L260-1. Fig 2b should be Fig 2c in the text. Unconditional abundance is shown in 2d-f which could be cited with each of a, b, c.

L272-281. Fig. 2 caption. There is redundancy in this caption. Delete ‘The first row of the figure shows’. This is not concise. It also seems to be at odds with the layout of the Fig panels which are labelled a-f.

Discussion

I think the discussion could be more concise.

I think the first section of the discussion could be subdivided further to focus the reader’s attention. Perhaps rather than the formal subheading used currently, name the key variable identified – e.g. Influence of … Then a subsequent subheading (around L334) could be the Influence of forest type.

L290-3. This sentence is redundant.

L298. Add Kavanagh & Wheeler (2004); Lindenmayer et al. (2004).

L313-15. It needs to be made clear in the study area description that the younger stand ages may contain some old growth elements. Perhaps because stand replacing fires may be patchy.

L317. There seems some repetition here.

L319-22. The point here about detection and forest age could be made more concise. I assume this point may be less relevant when you model prob of absence.

L326. I think it worth also citing reference 18 which first identified an influence of elevation on- greater glider occupancy or decline.

L331. Or the climate model predicted temperature less well than elevation.

L334. High not higher.

L337. I think Kavanagh & Stanton (1998) (Aust Zool 30:449) should be cited here in relation to paucity in alpine ash and at the highest elevations.

L338. This sentence could be deleted.

L347-9. This sentence should be placed at the end of the section.

The following reference should also be cited. Salmona, J., Dixon, K.M., and Banks, S. C. (2018). The effects of fire history on hollow-bearing tree abundance in montane and subalpine eucalypt forests in southeastern Australia. Forest Ecology and Management 428, 93–103.

Ross Goldingay

7. PLOS authors have the option to publish the peer review history of their article (what does this mean?). If published, this will include your full peer review and any attached files.

Reviewer #2: No

Reviewer #3: No

---

## [Author Response · Author response to Decision Letter 1]

28 Feb 2022

Professor David Lindenmayer

Fenner School of Environment and Society

The Australian National University

Canberra, ACT, 2601

28 February 2022

Re: Further revisions to PONE-D-21-30842R1: Elevation, disturbance, and forest type drive the occurrence of a specialist arboreal folivore

Dear Dr Angela Marie White, Ph.D,

Thankyou for your correspondence of 24 February 2022. We have now completed a further set of revisions to our manuscript, entitled: Elevation, disturbance, and forest type drive the occurrence of a specialist arboreal folivore that was submitted to PLOS One. 

 We have carefully considered and then responded to the comments from Reviewer #3. The changes we have made to the paper are outlined in a point-by-point response in the remainder of this letter. 

 We are most grateful to Professor Ross Goldingay (who was Reviewer #3) for his insightful comments. Addressing his suggested changes has helped further strengthen the manuscript. 

 We trust that this letter and the associated revised manuscript will be received favourably and look forward to hearing from you in the near future. 

Yours sincerely,

David Lindenmayer

(On behalf of all authors)

 

COMMENTS FROM REVIEWER #3

Comments from Reviewer #3

This is an interesting and worthwhile study. The authors have done a good job of revising the ms based on the reviews of previous referees. I think some further revision will enhance the quality and readability of this ms. Some new paragraphs seem redundant given other text that is presented. Elsewhere I found there is existing or new text that is not concise and should be deleted. In a few places I think some additional references could be cited to link with earlier research conducted on this species. These suggestions are documented below. 

Detailed comments and corrections 

The line numbers refer to the clean copy. 

Response from authors

We thank Reviewer #3 (Professor Goldingay) for their support of our study and also for his insightful comments. We have further revised the manuscript to address the comments made by Professor Goldingay. These changes have further strengthened the paper and we are grateful for the opportunity to revise the manuscript. 

Comments from Reviewer #3

Abstract 

L32-33. This sentence can be deleted ‘We excluded data collected in these forest types in further analysis.’ 

Response from authors

We have removed this sentence as suggested by Reviewer #3. 

Comments from Reviewer #3

Introduction 

L69. Add Kavanagh & Lambert (1990) Aust Wildl Res 17:285. 

Response from authors

We have added the citation as requested. 

Comments from Reviewer #3

L70. Add Kavanagh & Wheeler (2004). p.413-25. In ‘The Biology of Australian Possums and Gliders’. Add Lindenmayer, et al. (2004). Wildlife Research 31, 569. 

Response from authors

We have added these citations as requested. 

Comments from Reviewer #3

Methods 

L116. Glider surveys were conducted during Dec 2020 to May 2021. Why are temperature data described for 2019-20 and yet the temperature data used in an analysis was from 2015-19? 

Response from authors

We had to use available temperature and other weather data in our modelling; data at the time of our field surveys are currently still not available. 

Comments from Reviewer #3

L139. Table S1 describes 3 not 2 temperature variables. 

Response from authors

We have corrected this error. 

Comments from Reviewer #3

Is it meaningful to go to 2 decimal places with all of the variables? In Table S2 as well. 

Response from authors

We have rounded these values to a single decimal place. 

Comments from Reviewer #3

Change ‘No’ to ‘No.’ 

Response from authors

We have changed No to “number” in the tables to make interpretation easier for readers. 

Comments from Reviewer #3

L141 & 144. It would be helpful to know explicitly the period over which the number of days were estimated to avoid confusion (i.e. is it 1 Jan 2015 to 31 Dec 2019? i.e. 5 years not 4). 

Response from authors

It is the full calendar year for each year – this has now been clarified in the revised Methods section. 

Comments from Reviewer #3

L148. ‘that remain’ needs some qualification. 

Response from authors

We have removed the term “that remain” from the revised manuscript. 

Comments from Reviewer #3

L171. It would be helpful to the reader if you stated the benefits of modelling species’ absence rather than presence. 

Response from authors

We have modelled species absence in this case, but presence is the opposite side of the same coin. Essentially in the hurdle part of the modelling, the first stage of the work is to quantifying the factors that influence absence/presence as we have done here. 

Comments from Reviewer #3

L199. It’s not clear whether there is justification to build models with 4 covariates. 

Response from authors

We have data from 123 sites from which to construct our models. The number of potential explanatory variables should be at least an order of magnitude less than the number of sites – and we have elected to construct models with a relatively small number of covariates that are ecologically meaningful. 

Comments from Reviewer #3

L201-3. Awkward sentence structure. Perhaps put variables in brackets or simplify to be referred to as temperature variables. 

Response from authors

We have reworked the sentence (and broken it into two shorter sentences) so that it can be more readily understood by readers. 

Comments from Reviewer #3

L206-8. Do the levels in the variables need to be restated here? 

Response from authors

We have removed the repetition regarding the age classes from this part of the paper. 

Comments from Reviewer #3

L209-10. Given the high correlation between elevation and the 2 temperature variables it is not clear why all 3 should be retained. Two appear redundant. One temp variable certainly is. 

Response from authors

At the outset of the analysis, it was unclear which of the potential explanatory variables would be most important. On this basis, we implemented our analysis via four sets of models (each set with 8 models) – giving 32 models all up. Only one temperature variable was fit within any given set. This is carefully described in the revised version of the manuscript. 

Comments from Reviewer #3

L223-229. Delete. This seems to be covered more concisely at L201-3. Note at L223 ‘presence’ rather than ‘absence’ is used. 

Response from authors

We have revised this section of the paper as requested. 

Comments from Reviewer #3

There was a high correlation between the 2 temperature variables so only one should be used. 

Response from authors

As outlined above, at the outset of the analysis, it was unclear which of the potential explanatory variables would be most important. On this basis, we implemented our analysis via four sets of models (each set with 8 models) – giving 32 models all up. Only one temperature variable was fit within any given set. This is carefully described in the revised version of the manuscript. 

Comments from Reviewer #3

L229-236. Delete. This section seems redundant given it precedes the results section in which one could concisely state what is presented, and leave any discussion of the models to the discussion. 

Response from authors

We have removed the text in this part of the manuscript. 

Comments from Reviewer #3

Results 

L240-46. This could be written more concisely. Some text seems repetitious of what is stated elsewhere (e.g. confining analysis to only the mountain ash forest). Describe the greater glider findings first. 

Response from authors

We have reworked the text to make the writing more concise. 

Comments from Reviewer #3

L251. Delete ‘As outlined above,’ 

Response from authors

We have deleted this text. 

Comments from Reviewer #3

Table S3. It would help if the models were arranged from lowest to highest ∆LOOIC for one column. 

Response from authors

We trialled the process of re-ordering the models as requested by Reviewer #3. However, the way the models are presented reflects the sequential addition of covariates and increasing model complexity. To reorder the models by ∆LOOIC makes the sequence of model construction quite confusing and we believe that it is best to retain it the way it is now – and with the best fitting model highlighted in bold. 

Comments from Reviewer #3

L260-1. Fig 2b should be Fig 2c in the text. Unconditional abundance is shown in 2d-f which could be cited with each of a, b, c. 

Response from authors

We have checked the labelling of the Figures and corrected them as appropriate. 

Comments from Reviewer #3

L272-281. Fig. 2 caption. There is redundancy in this caption. Delete ‘The first row of the figure shows’. This is not concise. It also seems to be at odds with the layout of the Fig panels which are labelled a-f. 

Response from authors

We have revised the caption for Figure 2 as requested. 

Comments from Reviewer #3

Discussion 

I think the discussion could be more concise. 

Response from authors

We have now carefully checked and then revised the entire Discussion section so that it is more concisely written. 

Comments from Reviewer #3

I think the first section of the discussion could be subdivided further to focus the reader’s attention. Perhaps rather than the formal subheading used currently, name the key variable identified – e.g. Influence of … Then a subsequent subheading (around L334) could be the Influence of forest type. 

Response from authors

This was a good suggestion from Reviewer #3 and we have restructured the opening sections of the Discussion as requested. 

Comments from Reviewer #3

L290-3. This sentence is redundant. 

Response from authors

We have removed this sentence as requested. 

Comments from Reviewer #3

L298. Add Kavanagh & Wheeler (2004); Lindenmayer et al. (2004). 

Response from authors

We have added these citations to the revised manuscript. 

Comments from Reviewer #3

L313-15. It needs to be made clear in the study area description that the younger stand ages may contain some old growth elements. Perhaps because stand replacing fires may be patchy. 

Response from authors

This was an astute point and we have now included this extra detail in the revised Methods section of the paper. 

Comments from Reviewer #3

L317. There seems some repetition here. 

Response from authors

We have removed the repetition from this part of the paper. 

Comments from Reviewer #3

L319-22. The point here about detection and forest age could be made more concise. I assume this point may be less relevant when you model prob of absence. 

Response from authors

We have removed much of the text at the end of this section as suggested by Reviewer #3. 

Comments from Reviewer #3

L326. I think it worth also citing reference 18 which first identified an influence of elevation on- greater glider occupancy or decline. 

Response from authors

We have added this citation in this part of the paper. 

Comments from Reviewer #3

L331. Or the climate model predicted temperature less well than elevation. 

Response from authors

This is a good point and we have added this caveat to the revised text. 

Comments from Reviewer #3

L334. High not higher. 

Response from authors

We have corrected this typographical error. 

Comments from Reviewer #3

L337. I think Kavanagh & Stanton (1998) (Aust Zool 30:449) should be cited here in relation to paucity in alpine ash and at the highest elevations. 

Response from authors

We have added this citation in this part of the paper. 

Comments from Reviewer #3

L338. This sentence could be deleted. 

Response from authors

We have deleted this sentence as recommended. 

Comments from Reviewer #3

L347-9. This sentence should be placed at the end of the section. 

The following reference should also be cited. Salmona, J., Dixon, K.M., and Banks, S. C. (2018). The effects of fire history on hollow-bearing tree abundance in montane and subalpine eucalypt forests in southeastern Australia. Forest Ecology and Management 428, 93–103. 

Response from authors

We have moved the sentence and added the citation as recommended. 

Comments from Reviewer #3

Ross Goldingay

Response from authors

We again thank Professor Goldingay for his most helpful and insightful comments. 

---

## [Editor Report · Decision Letter 2]

11 Mar 2022

Elevation, disturbance, and forest type drive the occurrence of a specialist arboreal folivore

PONE-D-21-30842R2

Dear Dr. Lindenmayer,

We’re pleased to inform you that your manuscript has been judged scientifically suitable for publication and will be formally accepted for publication once it meets all outstanding technical requirements.

Kind regards,

Angela Marie White, Ph.D.

Academic Editor

PLOS ONE
---

## [Editor Report · Acceptance letter]

17 Mar 2022

PONE-D-21-30842R2 

Elevation, disturbance, and forest type drive the occurrence of a specialist arboreal folivore 

Dear Dr. Lindenmayer:

I'm pleased to inform you that your manuscript has been deemed suitable for publication in PLOS ONE. Congratulations! Your manuscript is now with our production department. 

Kind regards, 

on behalf of

Dr. Angela Marie White 

Academic Editor

PLOS ONE